# Inverse-designed diamond photonics

Constantin Dory [1], Dries Vercruysse[1,4], Ki Youl Yang [1,4], Neil V. Sapra[1], Alison E. Rugar[1], Shuo Sun[1], Daniil M. Lukin[1], Alexander Y. Piggott[1], Jingyuan L. Zhang[1], Marina Radulaski[1,2], Konstantinos G. Lagoudakis[1,3], Logan Su[1] & Jelena Vučković [1]

Diamond hosts optically active color centers with great promise in quantum computation, networking, and sensing. Realization of such applications is contingent upon the integration of color centers into photonic circuits. However, current diamond quantum optics experiments are restricted to single devices and few quantum emitters because fabrication constraints limit device functionalities, thus precluding color center integrated photonic circuits. In this work, we utilize inverse design methods to overcome constraints of cutting-edge diamond nanofabrication methods and fabricate compact and robust diamond devices with unique specifications. Our design method leverages advanced optimization techniques to search the full parameter space for fabricable device designs. We experimentally demonstrate inverse-designed photonic free-space interfaces as well as their scalable integration with two vastly different devices: classical photonic crystal cavities and inverse-designed waveguide-splitters. The multi-device integration capability and performance of our inverse-designed diamond platform represents a critical advancement toward integrated diamond quantum optical circuits.

[1] E. L. Ginzton Laboratory, Stanford University, Stanford, CA 94305, USA. [2] Present address: Electrical and Computer Engineering, University of California, Davis, CA 95616, USA. [3] Present address: Department of Physics, University of Strathclyde, Glasgow G4 0NG, UK. [4] These authors contributed equally: Dries Vercruysse, Ki Youl Yang. Correspondence and requests for materials should be addressed to C.D. (email: cdory@stanford.edu) or to J.V. (email: jela@stanford.edu)

Diamond has excellent material properties for quantum optics[1,2], optomechanics[3,4], and nonlinear optics[5]. Of particular interest is the variety of color centers that diamond hosts, some of which exhibit very long coherence times[1,6]. The development of diamond photonic circuits[7] has emerged as a promising route for implementing optical quantum networks[8–18], quantum computers[19–21], and quantum sensors[22,23]. However, a major challenge in diamond quantum photonics is the lack of high-quality thin films of diamond, as the production of electronic grade diamond can be achieved only in homoepitaxy, and thinning processes are not repeatable enough for photonic crystal cavity fabrication[24]. As a result, state-of-the-art diamond cavity quantum photonics relies on angled-etching of bulk diamond[24]. This technique naturally leads to triangular cross-sections with strongly constrained geometries, which limit device design and functionality. Recent developments in diamond processing based on quasi-isotropic etching[25–28] (see Supplementary Note 1 and Supplementary Figs. 1 and 2) allow the production of diamond membranes with rectangular cross-sections and variable dimensions from bulk diamond. Although rectangular cross-sections are a major step toward diamond integrated circuits, this fabrication technique comes with its own geometric constraints, such as limitations on the range of fabricable feature sizes, which originate from a strong correlation of the initial etch depth and undercut thin-film area. Traditional photonic designs that do not account for fabrication constraints are thus unable to take full advantage of this new fabrication technique.

In this work, we overcome these fabrication and design challenges by employing inverse design methods. In silicon nanophotonics these methods have recently attracted considerable attention for their efficient design of devices with superior performance over conventional designs[29]. This optimization technique searches through the full parameter space of fabricable devices, thereby arriving at solutions previously inaccessible to traditional design techniques[30]. We showcase the potential of inverse design techniques for diamond integrated circuits by designing and fabricating several devices: a compact vertical coupler, an essential component for large-scale quantum photonic systems, and a small circuit consisting of inverse-designed vertical couplers and waveguide-splitters acting as interfaces for two nanoresonators. Our inverse-designed vertical coupler adheres to the diamond fabrication constraints and outperforms commonly used free-space interfaces. The fabricated devices show excellent agreement with simulations in terms of both performance and yield. In the second example, we illustrate the integration of such a vertical coupler into a diamond photonic circuit consisting of two nanobeam resonators connected via inverse-designed waveguide-splitters—a configuration that could be used to entangle two quantum emitters embedded inside such resonators.

## Results

**Inverse design of diamond nanophotonic devices.** In photonics, grating couplers are frequently used as optical free-space interfaces[31–36]. To achieve high coupling efficiencies, such designs typically use asymmetry along the z-axis, e.g., through partial etches[31,32,36] or material stacks with varying refractive indices[32]. In diamond quantum photonics many of these approaches cannot be employed because current thin-film diamond on silica substrate platforms[33–35] do not support state-of-the-art quantum optics experiments[20,21,24]. Similar approaches with hybrid structures, such as gallium phosphide (GaP) membranes on diamond, offer a platform for efficient grating couplers[37]. However, the optical field is confined in the GaP membrane and consequently emitters in diamond couple only evanescently to the field.

A practical solution to these fabrication and design challenges are notches (Fig. 1a), which are a perturbation to a waveguide with ≈1% scattering efficiency[20]. In our work, we develop an inverse-designed vertical coupler (Fig. 1b) and use the notch for a baseline comparison. The couplers have a footprint of $1.0 \times 1.0$ μm$^2$ and couple directly to a 400 nm wide waveguide without a tapering section, assuring compactness. As shown in Fig. 1c, the simulated peak efficiencies of the coupler (red) and the notch (green) are ≈25% and ≈1%[20], respectively. Furthermore, we optimize the vertical coupler to couple the light between

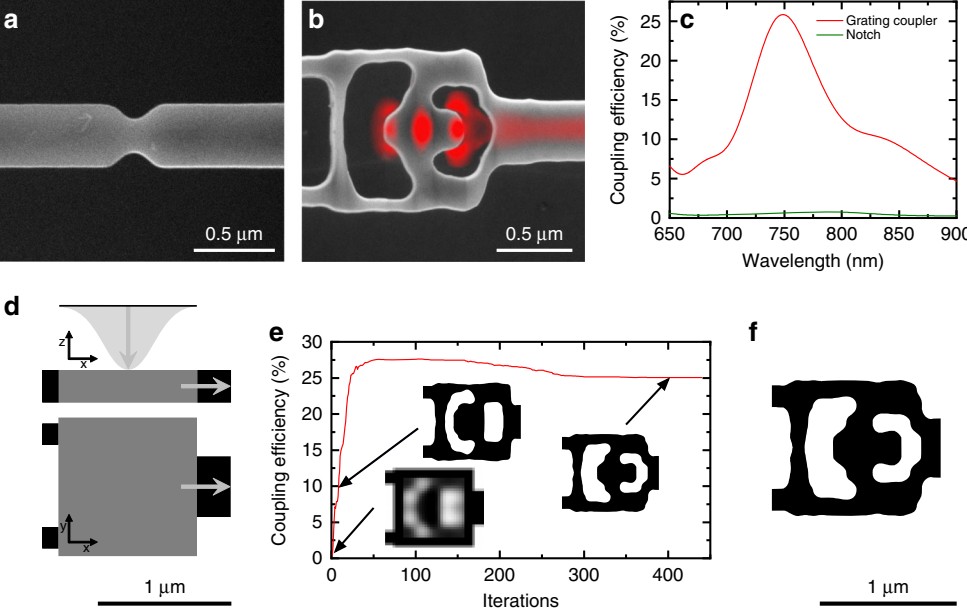

**Fig. 1** Inverse design of efficient nanophotonic interfaces. Scanning electron micrograph of **a** a notch and **b** an inverse-designed vertical coupler with simulated fields superimposed in red. **c** Simulated performance of the vertical coupler (red) and the notch (green). **d** Design set-up, where the gray area indicates the design area. **e** In-coupling efficiency during design optimization; insets illustrate different optimization phases. The small performance drop beyond 200 iterations of optimization occurs when fabrication constraints are imposed. **f** Final device design after optimization

the fundamental free-space mode $TEM_{00}$ and the TE fundamental mode of the waveguide. Even for conventional grating couplers in mature photonics platforms the selective coupling to the $TEM_{00}$ mode is a formidable challenge[31,32]. The theoretical maximum coupling efficiency of our couplers is 50%, because of the symmetry along the $z$-axis of our devices (i.e., the structure will couple light in $+/-$ $z$-direction equally).

Inverse design problems in photonics are defined by an electromagnetic simulation, a design region and a figure of merit to optimize. The starting conditions of the simulation for vertical couplers are shown in Fig. 1d. A vertically incident Gaussian beam forms the radiative source and is centered above the $1.0 \times 1.0$ $\mu m^2$ design region shown in gray. To the left of the design region are two black support bars to suspend the design, and to the right is a black output waveguide. The fraction of incident light coupled into the fundamental TE mode of the waveguide serves as the figure of merit and is maximized during our optimization process (detailed in ref. [38]). The coupling efficiency during the optimization is shown in Fig. 1e. At the start of the optimization, any permittivity value between that of air and diamond is allowed, which results in a continuous structure shown in the leftmost inset. After several iterations, this structure is discretized, in which case the permittivity is that of either air or diamond. This discrete structure is further optimized while also gradually imposing a penalty on infabricable features[39,40]. As a result, the coupling efficiency at a wavelength of 737 nm (silicon-vacancy color center zero-phonon line) peaks at a value of $\approx 27.5\%$, which then decreases to $\approx 25\%$ to comply with fabrication constraints[40].

## Characterization of diamond vertical couplers

To characterize the coupling efficiency of the vertical couplers, we measure the device shown in Fig. 2a, b, in top-down and sideview, respectively. An optical microscope image of the same structure, presented in Fig. 2c, qualitatively shows the high performance of

the vertical couplers. We characterize the polarization dependence of the vertical couplers by sweeping the polarization of the input laser beam (Fig. 2d). The observed fivefold reduction in the transmitted power when rotating the polarization by $\frac{\pi}{2}$ corresponds well to our simulated results (blue line in Fig. 2d) and is experimental evidence for excellent coupling to a linearly polarized $TEM_{00}$ mode. In Fig. 2e, we present experimentally determined efficiencies of the vertical couplers, which we acquire by coupling a tunable continuous-wave Ti:Sapphire laser to the structures in a cryostat using a 0.9 NA objective. We then collect the out-coupled beam with a single-mode polarization-maintaining fiber (PMF, black data points) and a multimode fiber (MMF, red data points). The experimental results show peak efficiencies of $\approx 21\%$ for PMF and $\approx 26.5\%$ for MMF, with broadband performance of >70 nm (PMF) and >90 nm (MMF). The small discrepancy between the measurements with PMF and MMF suggests that we couple very efficiently from the waveguide mode back into the fundamental free-space mode $TEM_{00}$. Moreover, the numerical simulation (blue line) agrees well with the experimental results.

Imposing fabrication constraints, such as minimum feature sizes, on the design optimization guarantees not only high fabrication yield but also robust performance, as we demonstrate in Fig. 2f. Here, we overlay transmission spectra of 15 different devices acquired with a supercontinuum source. During the experiments we purposely constrained ourselves to coarse alignment to confirm the robustness to alignment imperfections. The result of our analysis is shown in Fig. 2f, where the solid black line is the mean value of all couplers and the red shaded area indicates the standard deviation (SD) at a given wavelength. Moreover, the average efficiency of 30 devices fabricated with various doses is 24.2%.

## Diamond quantum optical interfaces

The vertical coupler presented in this work provides a compact, robust, and efficient

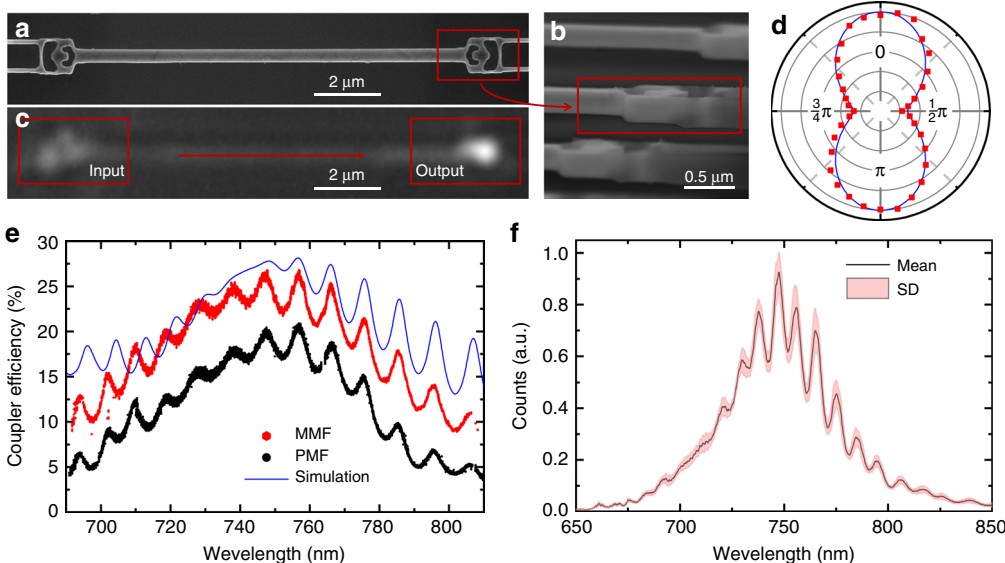

**Fig. 2** Inverse-designed vertical couplers. **a** Scanning electron micrograph of two vertical couplers connected by a waveguide. This device is used to characterize the efficiency of the vertical couplers. **b** Sideview of the vertical coupler, showing the undercut of the structure at an 85° angle. **c** Optical microscope image when focusing a Gaussian beam on the coupler on the left (input) and detecting the transmitted light from the coupler on the right (output). **d** Polarization scan shows that the vertical couplers preferentially couple to a Gaussian beam with a polarization perpendicular to the nanobeam. Simulated polarization dependence, shown as a blue line, are in good agreement with the measured data (red squares). **e** Efficiency of a single vertical coupler, peak efficiencies are $\approx 21\%$ for single-mode polarization-maintaining fibers (PMF, black) and $\approx 26.5\%$ for multimode fibers (MMF, red). Numerical simulation results are shown as a blue line. **f** Transmission spectra of 15 different devices using a supercontinuum source. The solid black line corresponds to the mean value and the red shaded area corresponds to the standard deviation

solution for free-space interfaces in cavity quantum electro-dynamics. In particular, our design is optimized to be compatible with simultaneous fabrication of high-$Q/V$ resonators for quantum optics experiments, as we avoid additional fabrication steps[31,32] that could impact the resonator performance. In this section, we therefore investigate coupling to the modes of nanophotonic resonators, which are used in quantum optics to enhance light-matter interactions[41] and to facilitate efficient integration of quantum emitters into optical circuits. We study nanobeam photonic crystal (PhC) cavities, which host a TE mode as shown in Fig. 3a, b. With a supercontinuum light source we acquire the transmission spectra shown in Fig. 3c by coupling a free-space laser beam into the TE fundamental mode of the nanobeam and subsequently into PhC modes. The data in red correspond to the device with vertical couplers, while the black spectrum corresponds to a cavity with notches as the free-space interface for the same input power and integration time. The count rates of the device with notches as an interface are more than two orders of magnitude smaller, for which we compensate by integrating ten times longer (data in green). When comparing the cavity resonances (blue arrows), we find a >550-fold increase in counts of the vertical coupler over the notch device for comparable quality factors ($Q \approx 4000$). This result matches well with the 625-fold enhancement that we expect from simulations. This improvement in coupling efficiencies allows for dramatically decreased experimental times (in some cases from weeks to minutes of photon integration), thereby opening opportunities for larger-scale experiments. In Fig. 3d we present spectra, where we couple the laser light directly to the cavity and optimize the alignment to collect maximum counts from the vertical coupler (red) and the notch (green). From this measurement, we can conclude that the extraction efficiency of light coupling from the cavity mode to the waveguide is ≈24 times greater for a vertical coupler than that for a notch, which corresponds well to the transmission experiment.

**Inverse-designed diamond photonic circuit**. For applications in quantum technologies, many nodes need to be connected to scale from single qubits to large, interconnected qubit arrays[8,10,18]. This requires the excitation of emitters in multiple cavities, the interference of their emission on beamsplitters, and subsequently the efficient collection and detection of photons. However, up until now elements such as waveguide-splitters have posed a major challenge in suspended diamond photonics, as state-of-the-art fabrication using angled etch is not conducive to variations in the device geometry. In contrast, as shown in Fig. 4a, we can fabricate a conceptual circuit comprised of three components with completely different geometries: vertical couplers, waveguide-splitters, and nanobeam PhC cavities. The device is designed to interfere the transmission of two nanobeam PhC cavities at an inverse-designed waveguide-splitter with a 50:50 splitting ratio and simulated efficiencies of 95%. We address the cavities separately or simultaneously by top-down excitation with a supercontinuum source focused on the cavities directly, as presented in Fig. 4b. The resonances of the two beams are detuned by <1 nm because of fabrication imperfections. We tune the two cavities into and out of resonance via gas condensation, as shown in Fig. 4c. Comparing the amplitudes of the cavity on and off resonance suggests constructive interference, indicating that the cavities are approximately in phase and have the same polarization. With this concept circuit, we show that inverse design can overcome limitations of classical photonics and enables large-scale on-chip quantum optics experiments. Extending this work, we can increase compactness by combining several functionalities into a single device, design circuits for arbitrary emitter locations, assure phase-matching across

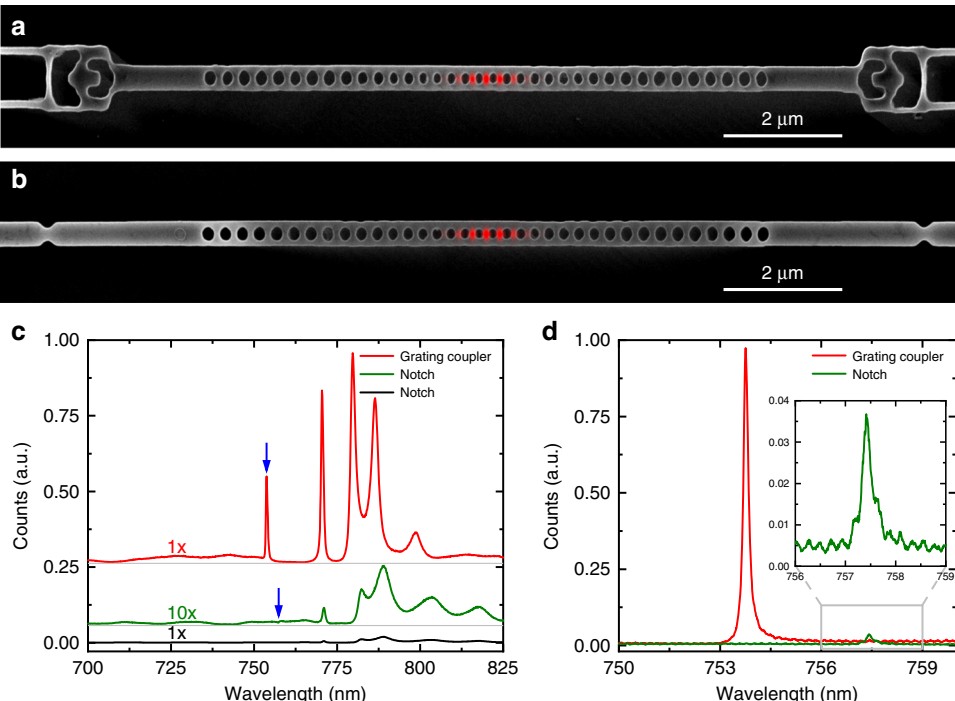

**Fig. 3** Suspended rectangular diamond nanobeams with optical interfaces. Scanning electron micrographs of nanobeam photonic crystal cavities in **a** with inverse-designed vertical couplers and in **b** with notches as interfaces for in- and out-coupling. Fields inside the cavities are depicted in red. **c** Transmission measurements using a supercontinuum light source. The red spectrum corresponds to the coupler device in **a**, black and green spectra to the notch device in **b**. The spectra are offset for better visualization and the cavity resonances are indicated by blue arrows. **d** Spectra acquired by coupling a supercontinuum light source directly to the cavity and out through a vertical coupler (red line) or a notch (green line). Inset corresponds to the data inside the gray box

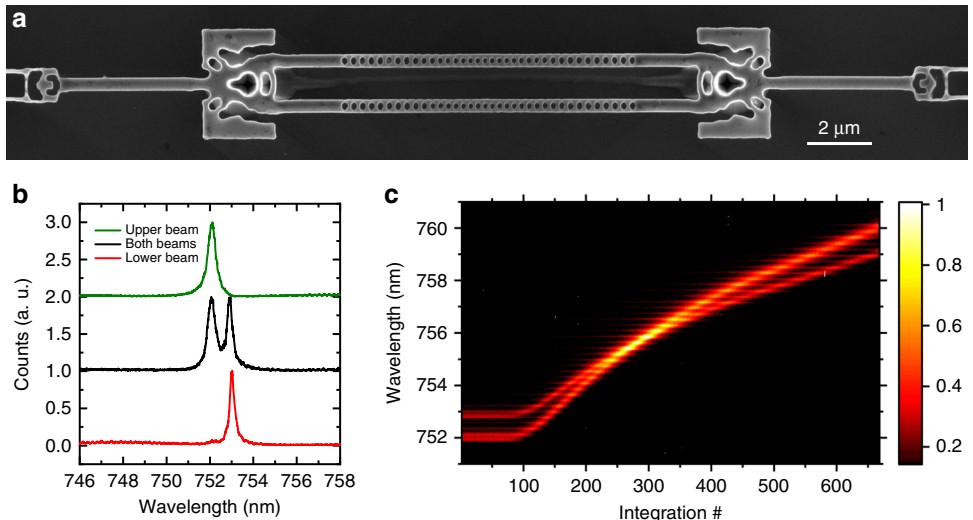

**Fig. 4** Interfacing grating couplers with a diamond photonic circuit. **a** Diamond photonic circuit, which could be used to entangle two emitters inside the two cavities. The circuit consists of a grating coupler, followed by a waveguide-splitter, and two resonators, the outputs of which are then recombined in a waveguide-splitter and coupled off-chip through a grating coupler. **b** Spectra of the nanobeams from the device shown in **a**. Green, black, and red data correspond to the upper, both and the lower nanobeam, respectively. **c** Demonstration that cavities with fabrication induced frequency offset can be tuned in resonance via gas tuning; colorbar corresponds to normalized counts

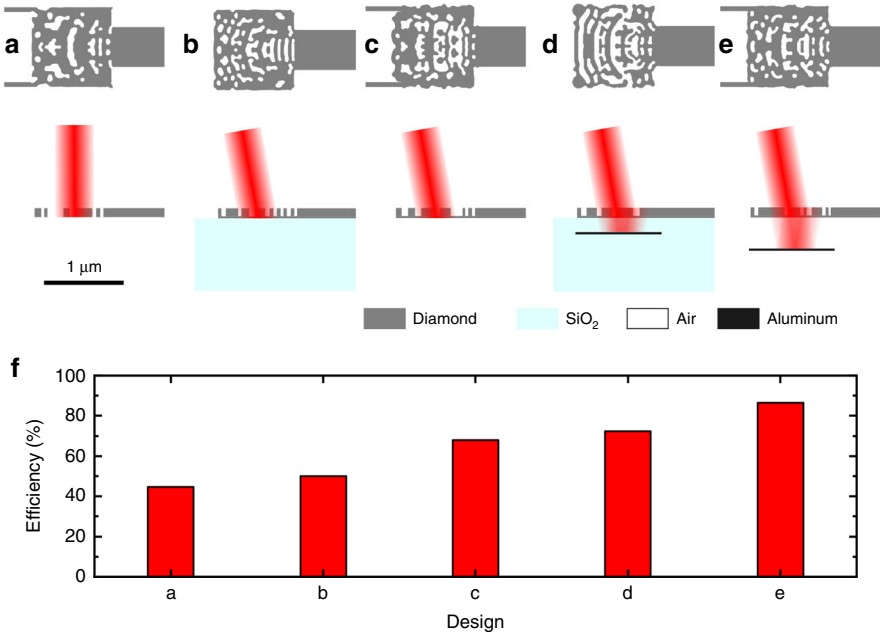

**Fig. 5** Designs for high-efficiency vertical couplers. **a** Vertically symmetric coupler suspended in air. Vertically asymmetric couplers employing partial etch with tilted incident laser beam (10°): vertical coupler **b** on SiO$_2$, **c** suspended in air, **d** on SiO$_2$ with aluminum back-reflector, and **e** suspended in air with aluminum back-reflector. **f** Simulated efficiencies of the devices shown in **a**–**e**: 44.7, 51.0, 67.9, 72.4, and 86.6%

different paths of the circuits, and optimize for specific bandwidths. Such a platform can then utilize the scalability that diamond color centers offer: site-controlled implantation of high-quality color centers[42,43] and small inhomogeneous broadening[44], which can be overcome by cavity-enhanced Raman emission[17,20] or strain tuning[45–48].

**Highly efficient free-space-waveguide interfaces.** Ultimately the implementation of scalable quantum networks requires efficiencies of building blocks close to unity. Efficiencies of >90% can be achieved with fiber tapers[49], which have the drawback of significantly larger footprints. To achieve comparable efficiencies,

we reduce the fabrication constraints to 60 nm feature sizes, increase the laser spot size, device footprint, and waveguide width. This allows us to improve the simulated efficiency to 44.7%. However, vertically symmetric devices, such as shown in Fig. 5a cannot exceed 50% efficiency. For further improvements, we tilt the incident laser beam by 10° and break the symmetry along the *z*-axis of the couplers via a partial etch[32]. In Fig. 5b, we show diamond devices on SiO$_2$ with efficiencies of 51.0%. Such devices could be achieved through diamond thin-film on SiO$_2$ production[5] or pick and place techniques[50] and are a promising route for a range of applications, including long-distance entanglement schemes, and nonlinear optics. Devices suspended in air

(Fig. 5c) have a larger refractive index contrast and show efficiencies of up to 67.9%. Additionally employing back-reflectors[31] as shown in Fig. 5d, e results in efficiencies of 72.4% and 86.6%, for diamond on $SiO_2$ and suspended structures, respectively. The back-reflector distance to the coupler (400 nm and 650 nm) is significantly shorter than the photon wave-packet and optimized to match the phase between reflected and directly coupled photons. These findings are encouraging for the development of highly efficient and compact photonic free-space interfaces as an alternative to fiber tapers for quantum photonic applications at the single-photon level. Moreover, many experiments will require optical driving of individual emitters to compensate for their spectral broadening via Raman processes[17,20]. This individual addressing is easier to implement in free-space coupling configurations than with many tapered fibers inside a cryostat. High efficiencies and compactness will be crucial in these experiments, as losses will be the limiting factor. Thus, inverse design is likely to play a major role in the development of such photonic circuits[30].

## Discussion

In summary, we employ optimization-based inverse design methods to overcome the constraints of cutting-edge diamond nanofabrication and to develop efficient building blocks for diamond nanophotonic circuits[51]. In optical free-space couplers and a small diamond photonic circuit we attain the crucial properties of high efficiency, compactness, and robustness. This work now enables more complex quantum circuits, where compact solutions for a variety of device components such as pulse shapers, splitter trees, phase delays[52], and mode converters[53] are critical. Thus, this progress lays the foundation for scaling to larger quantum networks[8,10,54] with spins[6] embedded in quantum nodes. In addition, inverse design methods can be applied to other promising material platforms that host quantum emitters and have challenging fabrication protocols, such as silicon carbide[55] and yttrium orthovanadate[56].

## Data availability

The data sets generated during and/or analyzed during this study are available from the corresponding authors on request.

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

## Acknowledgements

We acknowledge the help of Usha Raghuram and Elmer Enriquez with RIE. This work is financially supported by Army Research Office (ARO) (award no. W911NF1310309), Air Force Office of Scientific Research (AFOSR) MURI Center for Attojoule Nano-Optoelectronics (award no. FA9550-17-1-0002), National Science Foundation (NSF) Division Of Electrical, Communications Cyber Systems (ECCS) (award no. 1838976), and Gordon and Betty Moore Foundation; C.D. acknowledges support from the Andreas Bechtolsheim Stanford Graduate Fellowship and the Microsoft Research Ph.D Fellowship. K.Y.Y. and M.R. acknowledge support from the Nano- and Quantum Science and Engineering Postdoctoral Fellowship. D.M.L. acknowledges support from the Fong Stanford Graduate Fellowship. D.M.L. and A.E.R. acknowledge support from the National Defense Science and Engineering Graduate (NDSEG) Fellowship Program, sponsored by the Air Force Research Laboratory (AFRL), the Office of Naval Research (ONR) and the Army Research Office (ARO). D.V. acknowledges funding from FWO and European Unions Horizon 2020 research and innovation program under the Marie Sklodowska-Curie grant agreement No. 665501. We thank Google for providing computational resources on the Google Cloud Platform. Part of this work was performed at the Stanford Nanofabrication Facility (SNF) and the Stanford Nano Shared Facilities (SNSF), supported by the National Science Foundation under award ECCS-1542152.

## Author contributions

C.D. and J.V. conceived and designed the experiment. C.D. developed the fabrication techniques, fabricated the sample, and measured and analyzed the data. D.V., N.V.S., C.D., and L.S. conducted inverse design optimization of photonic components. K.Y.Y., D.M.L., and A.Y.P. contributed to the sample fabrication. A.E.R. and S.S. contributed to optical measurements. J.L.Z., M.R., and K.G.L. provided expertize. J.V. supervised the project. All authors participated in the discussion, understanding of the results, and the preparation of the manuscript.

## Additional information

**Competing interests:** The authors declare no competing interests.

