## [Peer Review File · Nature Communications]

Reviewers' Comments:

Reviewer #1:

Remarks to the Author:

In this paper, the authors apply inverse design to quasi-isotropic fabrication in diamond to demonstrate vertical grating couplers and the building blocks of a diamond nanophotonic circuit composed of two nanobeam cavities connected via waveguide splitters. While the utility of inverse design has been demonstrated many times previously with other materials (e.g. a large amount of work from this same group), it is nice to see this successfully applied to diamond, which is notoriously difficult to fabricate. This approach to fabrication allows for new types of photonic devices/circuits that would be significantly more challenging with e.g. suspended diamond photonics relying on angled etching.

While I think the work is promising and is of appropriate scope/rigor for publication in Nat Comm, I have a few comments/questions for the authors:

1) I find the title slightly misleading. Indeed the authors are optimizing diamond photonics circuits for quantum technology applications, but the use of "optimized" suggests that there's no room for further improvement. Perhaps that's being especially nitpicky as I realize this is optimized within the constraints imposed on the design process. Further, there is no demonstration of coupling to emitters or using this system to demonstrate anything one would typically describe as "quantum." While this system should allow for operation at the quantum level (similar to how elements like an optical fiber could be described as "quantum"), I'm not entirely convinced that this is an appropriate title for the results presented in this work.

2) The comparison to the results of the inverse-designed coupler to a notch coupler seems to be chosen to demonstrate the necessity of inverse design. While the inverse design will indeed find the optimal coupler given the fab constraints, the notch is not what one would start from if they had this fabrication technique available and were designing "by hand." From my understanding, the notch coupler used in e.g. ref 20 is chosen as a result of the constraints in the fabrication (i.e. difficulty in making a coupler such as in fig 1b using the angle etching). For the quasi-isotropic fabrication presented here, one could certainly envision more naive/traditional approaches to the grating coupler design that would significantly out perform the notch (i.e. a few period semi-circles).

3) What is the path toward achieving the theoretical max of 50% for this symmetric design? In the supplementary, the authors suggest that a decrease in minimum feature size would improve the efficiency. Could you give an example of the achievable efficiency would be if you for instance dropped the minimum feature size from 100nm to 75nm or 50 nm?

And for clarification on the commentary in the supplementary, is this limit due to the feature size relative to the wavelength of light in the material or relative to the spot size of the laser? If it's the latter, could you win at all by sacrificing the footprint and using a larger laser spot-size? While achieving compact devices is always desirable, an improvement to device coupling efficiency might be worth the tradeoff.

On a related note, the claims about lack of scalability of fiber taper architectures also seems a little harsh? For measurements involving interference of photons from two devices, improving from 25% to e.g. 80% collection efficiency available to fiber coupling represents a ten-fold improvement in success rate. Perhaps this is slightly naive thinking, but that means you would need 10x more devices to achieve the same overall e.g. entanglement rate, which would perhaps start to approach a similar overall foot print and would be a significantly more challenging measurement.

4) Are there any relatively straightforward fabrication paths to overcome the symmetry and achieve >50%? Or perhaps something like "pick and place" techniques or stamping techniques to

transfer to another substrate?

In the paper, the authors claim that current thin-film diamond on silica substrate platforms do not support state-of-the-art quantum optics experiments. It seems that citations 33-35 for thin-film diamond on silica are with NVs (or at 1550 without emitters) while the state of the art described by 20,21,24 are with SiVs. Could you elaborate a little further on this as to why these approaches are no-gos? Is it based on the properties of SiVs in these thin films? Of course, this inverse design approach could also be applied to optimize fabrication in that sort of architecture.

5) As discussed in the abstract, the utility of defects in diamond relies on their integration into photonic circuits, which is not really discussed/demonstrated in the main text. Perhaps these results are forthcoming, but do you have any further measurements of the properties of the emitters in these devices? The supplementary hints at emitter properties, but it's not clear what exactly you are showing in figure 5b. Is this an emitter in the cavity? Or in a waveguide? Or in the bulk material before/after fabrication? Clarification on the results presented would be appreciated.

Reviewer #2:

Remarks to the Author:

The current manuscript reports the fabrication of a new set of diamond nanophotonic components based on the inverse design. The photonic inverse design technique has been pioneered by the group the authors of this paper belong to. It is particularly powerful when fabrication constraints forbid high performance nanophotonic components by conventional design approaches. Diamond photonics, despite significant advances recently, remains a challenging device platform. The application of inverse design principles on this relatively exotic platform is thus justified and compelling. Specifically, the paper presents two inverse-designed diamond photonic components, a free-space coupler and a beam splitter, and integrated these components with photonic crystal cavities in a single device. The experimental results largely fulfill the promise by the inverse design, achieving near simulated free-space coupling efficiency and splitting ratios. The coupler demonstrated significantly enhanced performance compared to a notch design previously used, and show excellent repeatability among batches of devices. The last part of the paper showcasing two photonic crystal cavities combined via a beam splitter and coupled to free space is very impressive, projecting an outstanding prospect of this technology for building complex quantum photonic systems on diamond platform. The work represents a significant advancement in the field, and could enable next generation diamond devices that would be otherwise impossible with conventional techniques. Therefore, I recommend publication in Nature Communications once the following points are addressed.

My main question for the author is why the peak inverse-designed coupler efficiency is capped at 27%, far below the theoretical limit of 50%. Given that ~50% is routine for Si/oxide grating couplers, what additional factors limit the diamond coupler efficiency? If the incident light is allowed to come in at an angle, does the peak efficiency increase? It would be nice if some intuitions are discussed. In addition, I would suggest to tune down the comparison with notch scattering method, as the latter is not designed to give high efficiencies so the comparison appears a bit unfair. Arguably, the fiber taper scheme is still the best solution if high efficiency light-matter interface is critical. How to close the gap between current inverse design and the fiber taper method remains an interesting open question, and should be acknowledged.

Dear Editors and Reviewers,

We thank you for your thorough and positive assessment of our work. We are very pleased to read that both Reviewers recommend our manuscript for publication in *Nature Communications*. Reviewer 1 points out that *“it is nice to see this (inverse design) successfully applied to diamond, which is notoriously difficult to fabricate.”* Moreover, Reviewer 2 finds our progress to be promising for the diamond community: *“The last part of the paper showcasing two photonic crystal cavities combined via a beam splitter and coupled to free space is very impressive, projecting an outstanding prospect of this technology for building complex quantum photonic systems on diamond platform.”* In the following responses to the Reviewers’ comments and the revised manuscript, we have incorporated the suggestions of the Reviewers. Below we present the Reviewers’ comments in black and our responses in blue. We also highlight major changes in red and blue in the manuscript for the Reviewers’ convenience. After following the Reviewers’ suggestions, we believe our revised manuscript is improved thanks to the Reviewers’ feedback.

Reviewers' comments:

Reviewer #1 (Remarks to the Author):

In this paper, the authors apply inverse design to quasi-isotropic fabrication in diamond to demonstrate vertical grating couplers and the building blocks of a diamond nanophotonic circuit composed of two nanobeam cavities connected via waveguide splitters. While the utility of inverse design has been demonstrated many times previously with other materials (e.g. a large amount of work from this same group), it is nice to see this successfully applied to diamond, which is notoriously difficult to fabricate. This approach to fabrication allows for new types of photonic devices/circuits that would be significantly more challenging with e.g. suspended diamond photonics relying on angled etching.

While I think the work is promising and is of appropriate scope/rigor for publication in Nat Comm, I have a few comments/questions for the authors:

We thank the reviewer for his/her positive feedback and address the reviewer’s comments in detail below.

1) I find the title slightly misleading. Indeed the authors are optimizing diamond photonics circuits for quantum technology applications, but the use of “optimized” suggests that there’s no room for further improvement. Perhaps that’s being especially nitpicky as I realize this is optimized within the constraints imposed on the design process. Further, there is no demonstration of coupling to emitters or using this system to demonstrate anything one would typically describe as “quantum.” While this system should allow for operation at the quantum level (similar to how elements like an optical fiber could be described as “quantum”), i’m not entirely convinced that this is an appropriate title for the results presented in this work.

We appreciate the Referee's comment as we do not want to give the reader a wrong impression and changed the title to "Inverse-Designed Diamond Photonics".

2) The comparison to the results of the inverse-designed coupler to a notch coupler seems to be chosen to demonstrate the necessity of inverse design. While the inverse design will indeed find the optimal coupler given the fab constraints, the notch is not what one would start from if they had this fabrication technique available and were designing "by hand." From my understanding, the notch coupler used in e.g. ref 20 is chosen as a result of the constraints in the fabrication (i.e. difficulty in making a coupler such as in fig 1b using the angle etching). For the quasi-isotropic fabrication presented here, one could certainly envision more naive/traditional approaches to the grating coupler design that would significantly out perform the notch (i.e. a few period semi-circles).

We agree with the Referee that a grating coupler that outperforms the notch could be designed by hand. So we tried to emphasize in the article that the notch device is more of a 'baseline' for comparison rather than the best available free-space interface:

"[...] However, the optical field is confined in the GaP membrane and consequently emitters in diamond couple only evanescently to the field. ~~Because of these fabrication and design challenges, commonly used free-space interfaces in diamond quantum photonics are notches (Fig. 1a) with only ~1% scattering efficiency. This accumulates to a loss of ~99.99% in transmission experiments through a single cavity (involving two such couplers) and is indicative of the loss at any optically interfaced node in a quantum network. To improve on this, we develop an inverse-designed vertical coupler (Fig. 1b) that significantly outperforms the notch interface.~~ A practical solution to these fabrication and design challenges are notches (Fig. 1a), which are a perturbation to a waveguide with ~1 % scattering efficiency. In our work, we develop an inverse-designed vertical coupler (Fig. 1b) and use the notch for a baseline comparison. The couplers have a footprint of $1.0 \times 1.0 \mu\text{m}^2$ and couple directly to a 400 nm wide waveguide without a tapering section, assuring compactness. ~~It is important to note that the notch relies on the scattering of light due to a perturbation in the waveguide, whereas the vertical coupler presented here is optimized to couple the light into the fundamental free-space mode TEM_{00} . Even for conventional grating couplers in mature photonics platforms the selective coupling to the TEM_{00} mode is a formidable challenge. Furthermore, as our couplers are symmetric along the z-axis, they have a theoretical maximum coupling efficiency of 50%.~~ Furthermore, we optimize the vertical coupler to couple the light between the fundamental free-space mode TEM_{00} and the TE fundamental mode of the waveguide. Even for conventional grating couplers in mature photonics platforms the selective coupling to the TEM_{00} mode is a formidable challenge. The simulated peak efficiencies of the coupler (red) and the notch (green) are 25% and 1%, as shown in Fig. 1c. The theoretical maximum coupling efficiency of our couplers is 50%, because of the symmetry along the z-axis of our devices. [...]"

"[...] When comparing the cavity resonances (blue arrows), we find a >550-fold increase in counts of the vertical coupler over the notch device for comparable quality factors ($Q \sim 4000$). This result is comparable to the 625-fold enhancement that we expect from simulations ~~and is~~

equivalent to the expected increase in communication rates in quantum networks. This improvement in coupling efficiencies allows for dramatically decreased experimental times (in some cases from weeks to minutes of photon integration), thereby opening opportunities for larger-scale experiments. [...]"

3) What is the path toward achieving the theoretical max of 50% for this symmetric design? In the supplementary, the authors suggest that a decrease in minimum feature size would improve the efficiency. Could you give an example of the achievable efficiency would be if you for instance dropped the minimum feature size from 100nm to 75nm or 50 nm?

We designed a symmetric device with smaller feature sizes of 60 nm, a larger spot size and waveguide. This resulted in coupling efficiencies of ~45%. We included a discussion of these results into the manuscript and later in this response letter.

And for clarification on the commentary in the supplementary, is this limit due to the feature size relative to the wavelength of light in the material or relative to the spot size of the laser? If it's the latter, could you win at all by sacrificing the footprint and using a larger laser spot-size? While achieving compact devices is always desirable, an improvement to device coupling efficiency might be worth the tradeoff.

We thank the reviewer for pointing out that the comments in the supplementary material are unclear. The minimum feature size is limited by our fabrication process, but the required minimum features are dependent on the wavelength of the light (the shorter the wavelength, the smaller the features necessary). We designed the couplers based on the minimum spot size of our setup as well as the waveguide dimensions given by our cavity design. For this specific problem we did not see an improvement in coupling efficiency by increasing the device footprint. However, we find a larger laser spot-size and waveguide to be beneficial for coupling efficiencies. We rewrote the section of the supplementary material and added the missing information:

"In our simulations we designed a vertical coupler based on the minimum spot-size of our setup, as well as waveguide dimensions given by our cavity design. Thus, increasing the footprint of the vertical couplers did not show improvement in coupling efficiencies once the footprint was larger than the laser spot-size. We observed that as soon as the dimensions of the coupler exceed the spot size of the laser, the coupling efficiency does no longer increase. Thus we decided to work with $1.0 \times 1.0 \mu\text{m}^2$ vertical couplers. For the current designs we use conservative minimum feature sizes of 100 nm during the device optimization. A decrease in minimum feature size relative to the wavelength would improve the efficiency of the vertical couplers. Furthermore, increasing the laser spot-size and device footprint as well as waveguide dimensions results in significantly improved efficiencies, which allow us to approach the fundamental limit of 50% (as shown in Figure 5 of the main manuscript)."

On a related note, the claims about lack of scalability of fiber taper architectures also seems a little harsh? For measurements involving interference of photons from two devices, improving from 25% to e.g. 80% collection efficiency available to fiber coupling represents a ten-fold

improvement in success rate. Perhaps this is slightly naive thinking, but that means you would need 10x more devices to achieve the same overall e.g. entanglement rate, which would perhaps start to approach a similar overall footprint and would be a significantly more challenging measurement.

We apologize for the harsh language in the previous version of the manuscript, we agree that to date fiber tapers are significantly more efficient and changed the sentence as follows:

“Ultimately, the implementation of scalable quantum networks requires efficiencies of building blocks close to unity. Efficiencies of > 90 % can be achieved with fiber tapers,⁴¹ which have the drawback of significantly larger footprints.”

4) Are there any relatively straightforward fabrication paths to overcome the symmetry and achieve >50%? Or perhaps something like “pick and place” techniques or stamping techniques to transfer to another substrate?

We see angled incident light, partial etches, and the use of hybrid material stacks (e.g. through pick and place or stamping techniques) a promising route forward. To check this hypothesis, we ran simulations using partial etch and angled incident light and added them to the manuscript:

Figure 5: Designs for high efficiency couplers. (a) Vertically asymmetric couplers employing partial etch with tilted incident laser beam (10°): Vertical coupler (b) on SiO₂, (c) suspended in air, (d) on SiO₂ with aluminum back-reflector and (e) suspended in air with aluminum back-reflector. (f) Simulated efficiencies of the devices shown in (a)-(e): 44.7%, 51.0%, 67.9%, 72.4%, and 86.6 %.

“Ultimately, the implementation of scalable quantum networks requires efficiencies of building blocks close to unity. Efficiencies of > 90 % can be achieved with fiber tapers,⁴¹ which have the drawback of significantly larger footprints. To achieve comparable efficiencies, we reduce the fabrication constraints to 60 nm feature sizes, increase the laser spot-size, device footprint and waveguide width. This allows us to increase the simulated efficiency to 44.7 %. However,

vertically symmetric devices as shown in Fig. 5a cannot exceed 50 % efficiency. For further improvements, we tilt the incident laser beam by 10° and break the symmetry along the z-axis of the couplers via a partial etch.³² In Fig. 5b we show diamond devices on SiO₂ with efficiencies of 51.0 %. Such devices could be achieved through diamond thin-film production⁵ or pick and place techniques⁴² and are a promising route for a range of applications including long-distance entanglement schemes, and nonlinear optics. Devices suspended in air (Fig. 5c) have a larger refractive index contrast and show efficiencies of up to 67.9 %. Additionally employing back-reflectors³¹ as shown in Fig. 5d and e results in efficiencies of 72.4 % and 86.6 %, for diamond on SiO₂ and suspended structures, respectively. The back-reflector distance to the coupler (400 nm and 650 nm) is significantly shorter than the photon wave-packet and optimized to match the phase between reflected and directly coupled photons. These findings are encouraging for the development of highly efficient and compact photonic free-space interfaces as an alternative to fiber tapers for quantum photonic applications at the single-photon level. Moreover, many experiments will require optical driving of individual emitters to compensate for their spectral broadening via Raman processes.^{17, 20} This individual addressing is easier to implement in free-space coupling⁶ configurations than with many tapered fibers inside a cryostat. High efficiencies and compactness will be crucial in these experiments, as losses will be the limiting factor. Thus, inverse design is likely to play a major role in the development of such photonic circuits.”

In the paper, the authors claim that current thin-film diamond on silica substrate platforms do not support state-of-the-art quantum optics experiments. It seems that citations 33-35 for thin-film diamond on silica are with NVs (or at 1550 without emitters) while the state of the art described by 20,21,24 are with SiVs. Could you elaborate a little further on this as to why these approaches are no-gos? Is it based on the properties of SiVs in these thin films? Of course, this inverse design approach could also be applied to optimize fabrication in that sort of architecture.

We thank the reviewer for pointing out this oversight, changed the sentence to be more specific:

“[...] However, a major challenge in diamond quantum photonics is the lack of high-quality thin films of diamond, as the production of electronic grade diamond can only be achieved in homoepitaxy, and thinning processes are not repeatable enough for photonic crystal cavity fabrication.²⁴ As a result, state-of-the-art color center diamond cavity quantum photonics relies on angled-etching of bulk diamond.²⁴ This technique naturally leads to triangular crosssections with strongly constrained geometries, which limit device design and functionality. [...]”

Furthermore, the reviewer is correct that only electronic grade diamond can yield high quality SiV. However, we fully agree that our optimization techniques will work well with diamond thin film on oxide platforms and added simulated devices to the manuscript.

5) As discussed in the abstract, the utility of defects in diamond relies on their integration into photonic circuits, which is not really discussed/demonstrated in the main text. Perhaps these results are forthcoming, but do you have any further measurements of the properties of the emitters in these devices? The supplementary hints at emitter properties, but it's not clear what

exactly you are showing in figure 5b. Is this an emitter in the cavity? Or in a waveguide? Or in the bulk material before/after fabrication? Clarification on the results presented would be appreciated.

The measurements in the supplemental material were taken from an emitter in a waveguide, we added this missing information to the text:

“A typical photoluminescence spectrum of several SiV color centers in a waveguide at 4 K is shown in Fig. S5a.”

Unfortunately, we had a range of issues with diamond substrate supply and tool downtime, which did not allow us to perform further quantum optical studies on these systems, yet.

Reviewer #2 (Remarks to the Author):

The current manuscript reports the fabrication of a new set of diamond nanophotonic components based on the inverse design. The photonic inverse design technique has been pioneered by the group the authors of this paper belong to. It is particularly powerful when fabrication constraints forbid high performance nanophotonic components by conventional design approaches. Diamond photonics, despite significant advances recently, remains a challenging device platform. The application of inverse design principles on this relatively exotic platform is thus justified and compelling. Specifically, the paper presents two inverse-designed diamond photonic components, a free-space coupler and a beam splitter, and integrated these components with photonic crystal cavities in a single device. The experimental results largely fulfill the promise by the inverse design, achieving near simulated free-space coupling efficiency and splitting ratios. The coupler demonstrated significantly enhanced performance compared to a notch design previously used, and show excellent repeatability among batches of devices. The last part of the paper showcasing two photonic crystal cavities combined via a beam splitter and coupled to free space is very impressive, projecting an outstanding prospect of this technology for building complex quantum photonic systems on diamond platform. The work represents a significant advancement in the field, and could enable next generation diamond devices that would be otherwise impossible with conventional techniques. Therefore, I recommend publication in Nature Communications once the following points are addressed.

We thank the reviewer for the positive assessment of our work and address his/her comments in detail below.

My main question for the author is why the peak inverse-designed coupler efficiency is capped at 27%, far below the theoretical limit of 50%. Given that ~50% is routine for Si/oxide grating couplers, what additional factors limit the diamond coupler efficiency? If the incident light is allowed to come in at an angle, does the peak efficiency increase? It would be nice if some intuitions are discussed.

In the revised manuscript we added an additional Figure 5 with simulated devices, where we decrease minimum feature sizes, and increase the laser spot-size, design region and waveguide width. With this new symmetric design, we achieve efficiencies close to 50%.

In addition, I would suggest to tune down the comparison with notch scattering method, as the latter is not designed to give high efficiencies so the comparison appears a bit unfair.

We agree with the reviewer that the notch is not intended to be an efficient coupling element and rather a practical solution to allow for scattering via a perturbation of the waveguide. We tried to emphasize that the notch is rather a baseline for comparison than a state-of-the-art coupling device:

“[...] However, the optical field is confined in the GaP membrane and consequently emitters in diamond couple only evanescently to the field. ~~Because of these fabrication and design challenges, commonly used free-space interfaces in diamond quantum photonics are notches (Fig. 1a) with only ~1% scattering efficiency. This accumulates to a loss of ~99.99% in transmission experiments through a single cavity (involving two such couplers) and is indicative of the loss at any optically interfaced node in a quantum network. To improve on this, we develop an inverse-designed vertical coupler (Fig. 1b) that significantly outperforms the notch interface.~~ A practical solution to these fabrication and design challenges are notches (Fig. 1a), which are a perturbation to a waveguide with ~1 % scattering efficiency. In our work, we develop an inverse-designed vertical coupler (Fig. 1b) and use the notch for a baseline comparison. The couplers have a footprint of $1.0 \times 1.0 \mu\text{m}^2$ and couple directly to a 400 nm wide waveguide without a tapering section, assuring compactness. As shown in Fig. 1c, the simulated peak efficiencies of the coupler (red) and the notch (green) are 25% and 1%, respectively. It is important to note that the notch relies on the scattering of light due to a perturbation in the waveguide, whereas the vertical coupler presented here is optimized to couple the light into the fundamental free-space mode TEM_{00} . Even for conventional grating couplers in mature photonics platforms the selective coupling to the TEM_{00} mode is a formidable challenge. Furthermore, as our couplers are symmetric along the z-axis, they have a theoretical maximum coupling efficiency of 50%. Furthermore, we optimize the vertical coupler to couple the light between the fundamental free-space mode TEM_{00} and the TE fundamental mode of the waveguide. Even for conventional grating couplers in mature photonics platforms the selective coupling to the TEM_{00} mode is a formidable challenge. The theoretical maximum coupling efficiency of our couplers is 50%, because of the symmetry along the z-axis of our devices. [...]”

“[...] When comparing the cavity resonances (blue arrows), we find a >550-fold increase in counts of the vertical coupler over the notch device for comparable quality factors ($Q \sim 4000$). This result is comparable to the 625-fold enhancement that we expect from simulations ~~and is equivalent to the expected increase in communication rates in quantum networks.~~ This improvement in coupling efficiencies allows for dramatically decreased experimental times (in some cases from weeks to minutes of photon integration), thereby opening opportunities for larger-scale experiments. [...]”

Arguably, the fiber taper scheme is still the best solution if high efficiency light-matter interface is critical. How to close the gap between current inverse design and the fiber taper method remains an interesting open question, and should be acknowledged.

We agree that the fiber taper is the most efficient solution to date. We see ways to improve the coupling efficiencies by introducing angled-incident light, partial etches and back-reflectors. We added this discussion to the manuscript as follows:

Figure 5: Designs for high efficiency couplers. (a) Vertically asymmetric couplers employing partial etch with tilted incident laser beam (10°): Vertical coupler (b) on SiO_2 , (c) suspended in air, (d) on SiO_2 with aluminum back-reflector and (e) suspended in air with aluminum back-reflector. (f) Simulated efficiencies of the devices shown in (a)-(e): 44.7%, 51.0%, 67.9%, 72.4%, and 86.6%.

“Ultimately, the implementation of scalable quantum networks requires efficiencies of building blocks close to unity. Efficiencies of $> 90\%$ can be achieved with fiber tapers,⁴¹ which have the drawback of significantly larger footprints. To achieve comparable efficiencies, we reduce the fabrication constraints to 60 nm feature sizes, increase the laser spot-size, device footprint and waveguide width. This allows us to increase the simulated efficiency to 44.7%. However, vertically symmetric devices as shown in Fig. 5a cannot exceed 50% efficiency. For further improvements, we tilt the incident laser beam by 10° and break the symmetry along the z-axis of the couplers via a partial etch.³² In Fig. 5b we show diamond devices on SiO_2 with efficiencies of 51.0%. Such devices could be achieved through diamond thin-film production⁵ or pick and place techniques⁴² and are a promising route for a range of applications including long-distance entanglement schemes, and nonlinear optics. Devices suspended in air (Fig. 5c) have a larger refractive index contrast and show efficiencies of up to 67.9%. Additionally employing back-reflectors³¹ as shown in Fig. 5d and e results in efficiencies of 72.4% and 86.6%, for diamond on SiO_2 and suspended structures, respectively. The back-reflector distance to the coupler (400 nm and 650 nm) is significantly shorter than the photon wave-packet and optimized to match the phase between reflected and directly coupled photons. These findings are encouraging for the development of highly efficient and compact photonic free-space interfaces as an alternative to

fiber tapers for quantum photonic applications at the single-photon level. Moreover, many experiments will require optical driving of individual emitters to compensate for their spectral broadening via Raman processes.^{17, 20} This individual addressing is easier to implement in free-space coupling⁶ configurations than with many tapered fibers inside a cryostat. High efficiencies and compactness will be crucial in these experiments, as losses will be the limiting factor. Thus, inverse design is likely to play a major role in the development of such photonic circuits.”

Reviewers' Comments:

Reviewer #1:

Remarks to the Author:

The authors have thoroughly addressed the points raised in the previous round of review. Further, the authors have added in new text and simulations that helps clarify the results and prospects for this approach, which is greatly appreciated.

I have no further concerns and I am happy to recommend this current version for publication.

Reviewer #2:

Remarks to the Author:

The authors have addressed my previous critiques satisfactorily. Therefore I recommend publication in Nature communications.

We want to thank the Reviewers once again for their helpful reviews. No further responses are necessary.

REVIEWERS' COMMENTS:

Reviewer #1 (Remarks to the Author):

The authors have thoroughly addressed the points raised in the previous round of review. Further, the authors have added in new text and simulations that helps clarify the results and prospects for this approach, which is greatly appreciated.

I have no further concerns and I am happy to recommend this current version for publication.

Reviewer #2 (Remarks to the Author):

The authors have addressed my previous critiques satisfactorily. Therefore I recommend publication in Nature communications.